# A Method for Translating Automotive Body-Related CAN Messages Based on Labeled Bits

**Zixiang Bi** [1] , **Guosheng Xu** [1,*] , **Chenyu Wang** [1] , **Guoai Xu** [2] **and Sutao Zhang** [1]

1   School of Cyberspace Security, Beijing University of Posts and Telecommunications, Beijing 100876, China
2   School of Computer Science and Technology, Harbin Institute of Technology, Shenzhen 518055, China
*   Correspondence: guoshengxu@bupt.edu.cn

**Abstract:** Traditional mechanical parts have been increasingly replaced by in-vehicle electronic control units (ECUs) that communicate via control area networks (CAN). For security reasons, the Database CAN (DBC) file, which defines the contents of CAN messages, is deemed confidential by original equipment manufacturers (OEMs). However, confidentiality has severely hindered research on automotive intrusion detection systems (IDS) and automotive control network testing, which makes automotive aftermarket device development difficult. Previous research has used tokenization algorithms, machine learning algorithms, and diagnostic information to obtain coarse CAN message contents. However, there is a large gap between the results obtained with these methods and the information contained in DBC files. In order to reverse CAN messages in a fine-grained manner, we propose a method to reverse a body-related CAN message based on tagged bits. This method tags data bits by collecting CAN traffic in different vehicle states. The test messages are obtained by fuzzing the CAN messages based on the tagging results, and the candidate messages are obtained by combining them with the results of a packet analysis. The final reverse result was based on the column AND bit-by-bit of the candidate messages. The reverse results showed that the method proposed in this paper could accurately locate the bits representing or controlling the body behavior with high reverse accuracy.

**Keywords:** electronic control units; controller area network; database CAN; reverse; fuzzing; AND bit-by-bit

## 1. Introduction

Along with the increases in the number and computing power of onboard electronic control units (ECUsciteECURise), ECUs are replacing more and more vehicle components that were previously controlled by mechanical systems. Various vehicular functions, such as the powertrain control and the switching of body equipment, are controlled by different ECUs. In addition to performing the specified functions, ECUs must transfer data quickly and reliably to other ECUs. The control area network (CAN) bus has become the most widely used standard communication protocol in current in-vehicle networks due to its low cost, efficient propagation, and stability advantages [1]. However, the lack of security mechanisms included in the CAN protocol means that the confidentiality and integrity of CAN messages cannot be guaranteed [2], thereby exposing the control system in the vehicle to greater risk in terms of its security. Numerous studies have proven that the CAN bus can be attacked, and the consequences can be severe. For example, at the 2015 Black Hat and DEFCON conferences, Charlie Miller and Chris Valasek demonstrated the ease of controlling the steering wheel, brakes, lights, and turn signals of a JEEP Cherokee by hacking into the infotainment system connected to the CAN bus and then writing attack commands to the CAN bus [3]. In 2021, two researchers from Kunnamon used a drone to attack a Tesla car, open its doors, and change the steering and acceleration patterns through the injection of malicious CAN messages [4]. The consequences of the injection of malicious messages into the CAN bus by the attackers were catastrophic.

To prevent the malicious control of ECUs on the CAN network, original equipment manufacturers (OEMs) attempt to hide the database CAN (DBC) [5] files that describe CAN message definitions to prevent attackers from sending malicious control commands to the CAN network. Although this measure can increase the cost of attacks to some extent, it hinders the development of the automotive security research field and the automotive aftermarket. Most affected is research on intrusion detection systems for CAN networks. The proposed CAN intrusion detection techniques are based on the use of message transfer characteristics to detect anomalies and are practically independent of the behavior and state of the vehicle [6–9], which leads to limitations in the efficiency of existing solutions. In addition, security researchers usually use fuzzy test automation to test the security of communication protocols [2,10]. However, the lack of DBC files defining the CAN message format leads to inefficient and blind fuzzy tests against the CAN bus [11–13]. Moreover, the privatization of DBC files leads to ineffective access to vehicle status data for devices in the automotive aftermarket, which renders automotive driver assistance devices and status display tools meaningless.

In order to obtain a more effective basis for CAN bus intrusion detection and fuzz testing, research in the security field is also trying to obtain a clear definition of CAN messages. Initial research focused on the tagging of CAN message data types, mainly through FBCA [14], READ [15], and LibreCAN [16], which were limited to the classification of data according to its variability. Such methods cannot obtain vehicle-specific data. Later research attempted to identify vehicle behavior in CAN traces using machine learning algorithms [17,18], the efficiency of which depends on the coverage of the training data set. However, the application of these methods is limited by the fact that each vehicle has a unique CAN message definition. Some other researchers have tried to obtain a description of the vehicle state with CAN messages through the Unified Diagnostic Services (UDS) protocol [19,20]. However, these studies relied on vehicle state data supported by the UDS protocol, causing the final results to have limitations.

In recent work, we used cheap sensor devices to accurately obtain the bits describing vehicle driving data in CAN messages [21]. Since CAN traffic contains body-state-related data such as lights, doors, and seat belts in addition to vehicle driving data, CAN messages describing these data were not effectively obtained. In order to accurately translate the CAN messages describing the body state, this paper proposes an innovative CAN message translation method based on labeled bits. This method uses different car states to tag each bit in CAN messages and filter them to get the candidate messages related to the body state. For the candidate messages, using strategic fuzzing and packet analysis techniques, the body event described in the CAN message can be translated, and the bits describing the event can be located.

The contributions of this study can be summarized as follows:

- The reverse method described in this paper labels each bit of the data field corresponding to each ID based on the vehicle state and applies the idea of fuzzy testing to the production of test messages. It effectively filters the messages unrelated to the car body and quickly locates the candidate messages with improved efficiency.
- The innovative method proposed in this paper innovatively uses column AND bit-by-bit to locate useful bits, which improves the accuracy of the reverse method while reducing the number of dependent candidate messages.
- The method proposed in this paper can be applied to most cars, regardless of the vehicle model and make. Since commercially available vehicles must be equipped with a standard CAN data interface, our method only uses this interface for data transmission and reception.

The rest of the paper is organized as follows: In Section 2, we provide background knowledge and present related work. Section 3 details the steps taken to reverse CAN message based on tagged bits. Section 4 details experiments conducted on actual vehicles and analyze the accuracy, advantages, and efficiency of the approach used in this paper. The last section concludes the paper and gives some future prospects.

## 2. Background and Related Work

### 2.1. Controller Area Network(CAN)

The composition and format of a standard CAN bus data frame are shown in Figure 1. The structure consists of seven segments: frame start, arbitration segment, control segment, data segment, cyclic redundancy check (CRC) segment, acknowledge character (ACK) segment, and frame end. The two segments that are most relevant to this paper are the arbitration and data segments.

| Frame Structure | Frame Start | Arbitration Segment | | Control Segment | | | Data Segment | CRC Segment | | ACK Segment | | Frame End |
|---|---|---|---|---|---|---|---|---|---|---|---|---|
| Standard Frame | SOF | ID | RTR | IDE | R0 | DLC | Data Field | CRC | CRC delimiter | ACK | ACK delimiter | EOF |
| Number of Bits | 1 | 11 | 1 | 1 | 1 | 4 | 0-64 | 15 | 1 | 1 | 1 | 7 |

**Figure 1.** Structure of CAN frame.

The arbitration segment consists of 12 bits, the last of which is the remote transmission request (RTR), which is used to distinguish whether a frame is a data frame or a remote frame. The first 11 bits form the CAN ID, which identifies this CAN message's sending node (ECU). The bus can only have one ECU transmitting at a time, and each ECU can initiate message transmission. However, the bus does not specify the priority levels of the ECUs. The CAN bus determines the priority levels of the messages using the CAN ID, where the smaller the CAN ID is, the higher its priority, which is the arbitration mechanism.

The data segment is the entire data field of the message and is 8 bytes in size. This short frame structure gives the CAN a high degree of real-time and interference immunity. The data field contains the control commands or status data sent by the ECU, thus enabling the different ECUs to work together to ensure the proper operation of the vehicle. In practice, the data fields are defined differently for each ID. For example, bits 9 to 12 of the data field corresponding to the message ID 0x100 are defined as data related to the lights, while bits 9 to 12 of the data field corresponding to the message ID 0x120 are related to the doors. These definitions are determined entirely at the discretion of the vehicle manufacturer and are non-public. The method described in this paper aims to quickly obtain the definition of the body state in the CAN message.

### 2.2. Fuzzy Test

Fuzzy testing is a classic information security testing technique [22,23] that is often used to detect security vulnerabilities in software or systems. The purpose of fuzzy testing is to generate a large amount of data in a random or semi-random way and send it to the system being tested to determine whether there is a potential security vulnerability by monitoring the system's abnormal response. The simplest fuzzy testing method inputs completely random data to the target program, but this type of method is inefficient. The current mainstream fuzzy testing methods use existing data variants and input data modeling to generate test data. Such methods can quickly and effectively discover vulnerabilities. In this paper, fuzzy testing is not used for detecting security vulnerabilities but for finding bits in CAN messages that describe the behavior of a vehicle body. We use the idea of fuzzing to fuzz the candidate bits in the message and send them to the automotive CAN network, recording the fuzzy messages with responses for bit locations.

### 2.3. Related Works

Whether monitoring abnormal CAN messages, studying the vehicle's operating status, or implicitly authenticating the driver, it is essential to have comprehensive and accurate information on each bit of the CAN message's data field. This is done through translation. Currently, there are three types of reverse methods that can be used for the CAN message's data field.

The first method type is based on the data variation characteristics of each bit. For example, Markovitz et al. extracted the signals and their boundaries by observing the data variation pattern of each bit of the data field corresponding to the same ID over time [14]. Mirco Marchetti and Dario Stabili proposed an algorithm to calculate the bit-flip rate and the magnitude array for each bit of the data field for each ID in order to determine the data field signal boundaries [15]. Mert D. Pesé et al. optimized the field segmentation algorithm based on the study by [15]. They matched the correlations between the segments with the help of OBD-II diagnostic and smartphone sensor data, which led to a more accurate classification of the field segmentation [16]. However, these methods only classify at the segment level and are not specific to each bit. Moreover, they do not form a mapping of the segments in response to specific events, which makes the results less valuable.

The second approach is implemented based on diagnostic protocols. For example, Tae Un Kanget al. located the exact bits by matching the return value of the OBD-II diagnostic protocol with the data in the CAN message [24]. Bram Blaauwendraad et al. improved on the study conducted by [24]. They made two data structures by simultaneously using the results obtained with OBD-II diagnostics and the CAN bus data. They used the Pearson correlation coefficient to determine the correlations and thus locate the valuable bits [20]. Miki E. Verma et al. used algorithms to extract CAN message signals and label them with the help of OBD-II PIDs, thus enabling mapping from bits to ECU functions [25]. However, these methods are limited by what the diagnostic protocol can provide. The diagnostic protocol can only provide a small amount of information about the motion status and very little about the vehicle body.

The third method type is implemented with the help of machine learning techniques. For example, Alessio Buscemi et al. and Clinton Young et al. identified multiple CAN message features as being needed for machine learning and trained their models with them, thus making their models capable of classifying CAN messages [17,26]. However, the CAN message features used to train machine learning models need to be defined manually, and the number of classification categories they could eventually identify is limited.

In our previous work, differential-based [27] and linear-regression-based [21] CAN message reversal methods have been successively proposed. The first proposed differential-based scheme finds the control bits of the car operation by performing a differential on the CAN datasets collected when the car is stationary and dynamic. Although this approach enables bit-level CAN message reverse, it is inefficient and inaccurate. In 2022, a multiple-linear-regression-based reverse framework was proposed, which built a multiple linear regression model between sensor data and CAN message data bits and used the model parameters to locate the bits controlling the vehicle's behavior quickly. This approach allows for the inversion of vehicle behaviors with higher efficiency and accuracy by only collecting data once. However, as the sensor data only reflects the behavior related to the vehicle driving or performing an operation (e.g., vehicle speed, acceleration, pedal angle), it causes this method to fail for CAN messages describing the state of the vehicle (e.g., lights and doors). The complete DBC file defines the content of each valid message in the CAN network. In order to fill the gaps in the previous work and reverse the status CAN message, this paper proposes a translation method based on labeled bits. The combined effect of this paper's solution and the previous work allows us to achieve results close to the DBC file.

## 3. Methodology

By analyzing related work and the current research status of the automotive cybersecurity field, we concluded that the research field and the automotive parts market are eager to achieve technological and product breakthroughs related to DBC files. In the current research, the reverse results obtained with existing methods were far from the contents of DBC files, except when our team's bit-level reverse method using multiple linear regression models was employed. However, our previous work focused on the reversal of CAN traces with messages related to car driving behaviors. Although the results were very similar to

the information contained in the DBC files, the body behaviors defined in the DBC files were still blank, for example, the information on lights, wipers, and doors.

In order to fill the gaps in the previous work, this paper proposes an innovative technique to reverse body-related CAN messages based on labeled bits. This technique first tags and segments the CAN traces collected in different scenarios and filters out the body-related candidate messages based on tags related to the body behavior. After that, the valid messages corresponding to the automotive body behaviors are recorded using fuzzing and packet analysis techniques. Finally, the valid messages corresponding to each body behavior are executed AND OPERATION bit-by-bit to get the valid bits describing the automotive body behavior in CAN messages. The overall flow of the method described in this paper is shown in Figure 2.

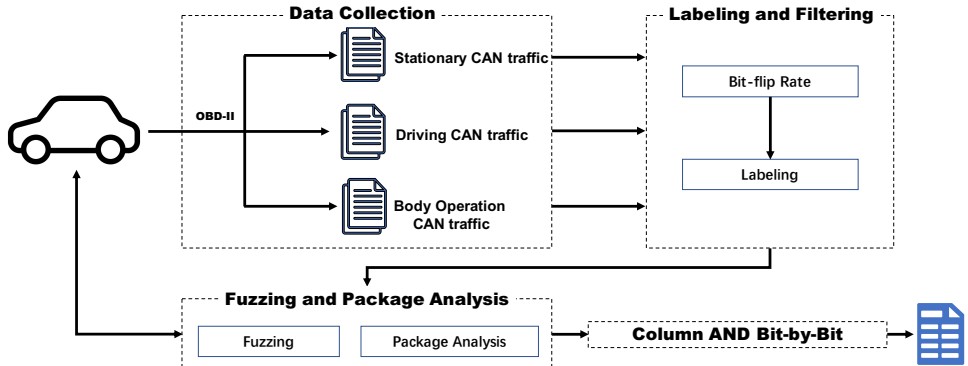

**Figure 2.** Overall flow of the translating method.

### 3.1. Data Collection

In the data collection phase, the approach described in this paper uses a combo cable to acquire the CAN traces of the target vehicle, as shown in Figure 3a. This combo cable consists of an OBD-II to DB9 diagnostic cable and a PCAN-USB FD adapter [28]. As shown in Figure 3b, the cable connects the vehicle's OBD-II port to a USB port on the computer side, which allows the computer to read and write CAN bus information to enable the experiment to proceed. It is not necessary to use the entire CAN message in our approach. Only the message ID, data segment, and timestamp must be captured. In this phase, the raw data include three data sets: the stationary ($\Omega_s$), the motion ($\Omega_m$), and the body operation ($\Omega_b$) data sets. To avoid blindness when collecting data, we previously analyzed a portion of the available DBC files and summarized the possible car body behavior described by CAN messages. The results are shown in Table 1.

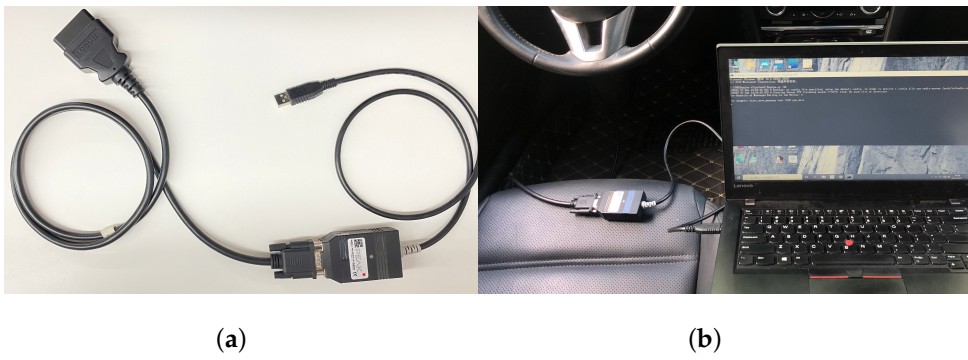

(**a**)             (**b**)

**Figure 3.** Data collection equipment and environment. (**a**) Combination cable for data collection; (**b**) In-vehicle data collection environment.

- $\Omega_s$ records the CAN traces without operation when the car is stationary after ignition;
- $\Omega_m$ contains the CAN messages of the driver during everyday driving of the car on a regular road;
- $\Omega_b$ records CAN traffic during the execution of body operations in place after the car is started. The operations are shown in Table 1.

**Table 1.** Vehicle-body-related functions and operations.

| Automotive Features | Body Operation |
|---|---|
| Light | Hazard Lights, Left Turn Signal, Right Turn Signal, Low Beam Light, High Beam Light, Fog Light, Emergency Flasher |
| Wiper | Low Windshield Wipers, High Windshield Wipers |
| Pedal | Brake Pedal, Accelerator Pedal |
| Gear | Gear |
| Seat Belt | Driver's Seat Belt, Passenger's Seat Belt |
| Door | Driver's Door, Passenger's Door, Left Back Door, Right Back Door, All Doors Closed, Trunk |

### 3.2. Labeling and Filtering

In this phase, the bit-flip rate is used to label the messages corresponding to the IDs appearing in the CAN traces and filter the candidate messages based on the labeling results. The main flow is shown in Figure 4.

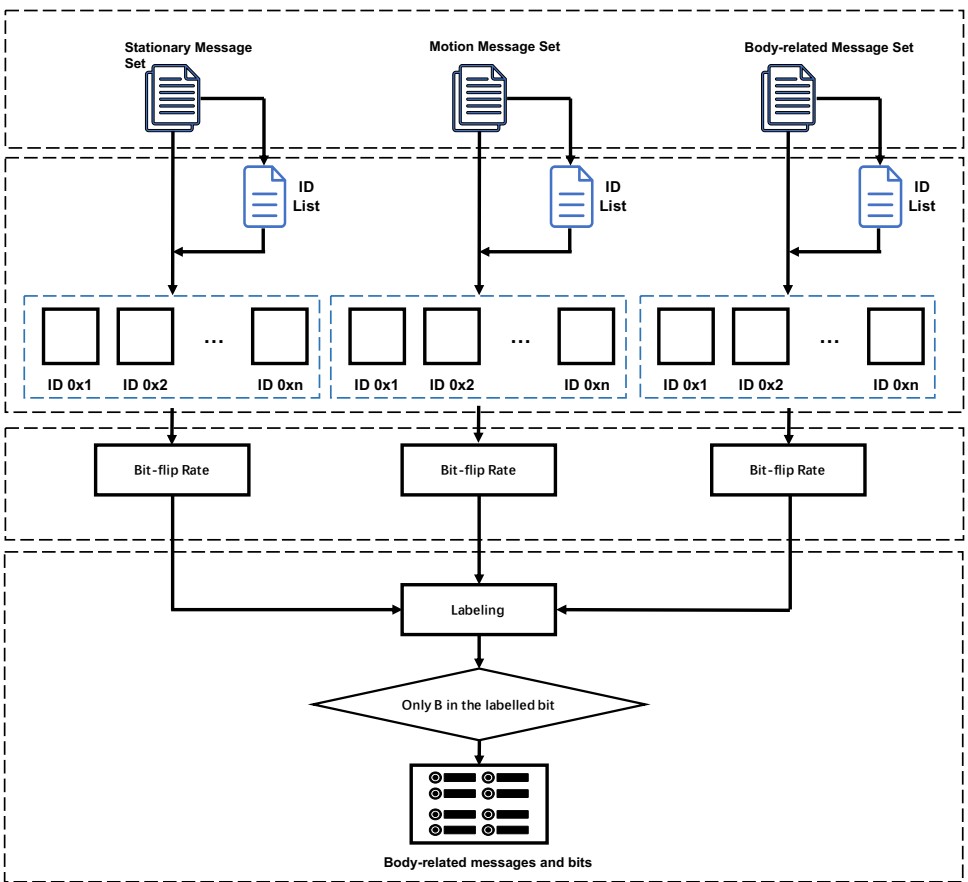

**Figure 4.** Labeling and filtering process.

First, the valid IDs are extracted from each dataset, and the messages are grouped according to the IDs. Then, for each dataset grouped by ID, the message data fields are sorted by timestamps. After that, to quickly and efficiently filter CAN messages for body-

related purposes, the bit flip rate is calculated for each message corresponding to each ID, and each bit is labeled based on the results of this calculation.

Calculating the bit flip rate is the most intuitive way to characterize the change in each bit. First, Equation (1) is used to calculate the number of flips of the $j$th bit in the message with ID $n$ ($BFN_n$, $j$), and the standard CAN protocol with a bit length of 64 bits is taken as an example, where $|id_n|$ indicates the number of messages with ID $n$ and $V_{n,i,j}$ indicates the value of the $j$th bit in the $i$-th message with ID $n$. After that, Equation (2) is used to get the flip rate $BFR_{n,j}$ of the $j$th bit in the message with ID $n$. By traversing all bits in each legal ID in the CAN traffic, the flip rate of each bit in each message can be obtained. In this step, we perform the bit flip rate calculation operation for each ID appearing in the three datasets ($\Omega_s$, $\Omega_m$, $\Omega_b$) to get the flip rate of each ID corresponding to the message.

$$BFN_{n,j} = \sum_{i=1}^{|id_n|-1} 1 \quad if \quad V_{n,i,j} \neq V_{n,i+1,j} \tag{1}$$

$$BFR_{n,j} = \frac{BFN_{n,j}}{|id_n|} \tag{2}$$

After that, the data bits of each ID is labeled based on the result of the bit flip rate calculation in the three datasets. Table 2 shows the mapping between the labels of the bits corresponding to each ID and the results of the bit-flip rate calculation in each dataset. First, the bits in all three datasets with a bit flip rate of 0 are labeled "None". Second, the label is kept consistent with the dataset for the bits with a non-zero bit flip rate in only one of the datasets. In addition, bits with non-zero results in any two data sets are labeled "Double". Finally, the bits with non-zero results in all three datasets are labeled "All". Figure 5 shows the process and outcomes of label mapping for the bit-flip rate inversion of the message with the ID 0x225.

**Table 2.** Mapping of bit flip rate results to labels.

| Bit-Flip Calculation Results | Label Results |
|---|---|
| Bit-flip rate is 0 in all datasets | None |
| Non-zero results exist only in $\Omega_s$ | S |
| Non-zero results exist only in $\Omega_m$ | M |
| Non-zero results exist only in $\Omega_b$ | B |
| Non-zero results in only two datasets | Double |
| Non-zero results in all datasets | All |

**Figure 5.** Examples of Labeling.

Based on the labeling results, our method can more quickly filter out messages related to car body behavior and the bits in the message data fields that control or represent the body state than other methods. The analysis of the tags revealed that bits with only B tags are most likely to handle/represent the state of the car body and can be directly used as valid bits in the candidate messages. The bits with other labels are related to driving, the stationary state, CTR, and checksum.

After this process, the CAN messages related to the body and the bits in these messages describing the body behavior are filtered out. This data can be further used as the basis to reverse further and determine how the body behavior is portrayed in the CAN messages.

### 3.3. Fuzzing and Package Analysis

Based on the labeling and filtering results, we determined which IDs correspond to data fields containing bits with potential for body behavior. In this phase, fuzzing and packet analysis are used to collect candidate messages that can trigger a vehicle body response.

First, messages with data fields containing bits labeled B are used as base messages. The approach described in this paper utilizes the idea of fuzzy testing by fuzzifying the bits marked B in the base message. In contrast, the non-candidate bits are left unchanged from the original data, thus allowing the test message to be obtained. An example of test message generation is given in Figure 6. First, based on the tagging and message filtering results, the ID corresponding to the message with the B tag in the data field is found. A real CAN message corresponding to this ID is randomly selected from the obtained messages. Afterward, the bits marked with B are fuzzed (randomized or traversed) according to the message marking results. Finally, the bits not marked with B are left unchanged, and a set of test messages corresponding to this ID is generated. Once the set of test messages has been obtained, the test messages are sent sequentially to the vehicle's CAN network via the OBD-II interface using the same cable combinations as used in the data collection phase. After each test message has been sent, we record each message that triggers the car's behavior and the body response it triggers. After this operation, we can obtain candidate messages that can trigger a specific response from the vehicle.

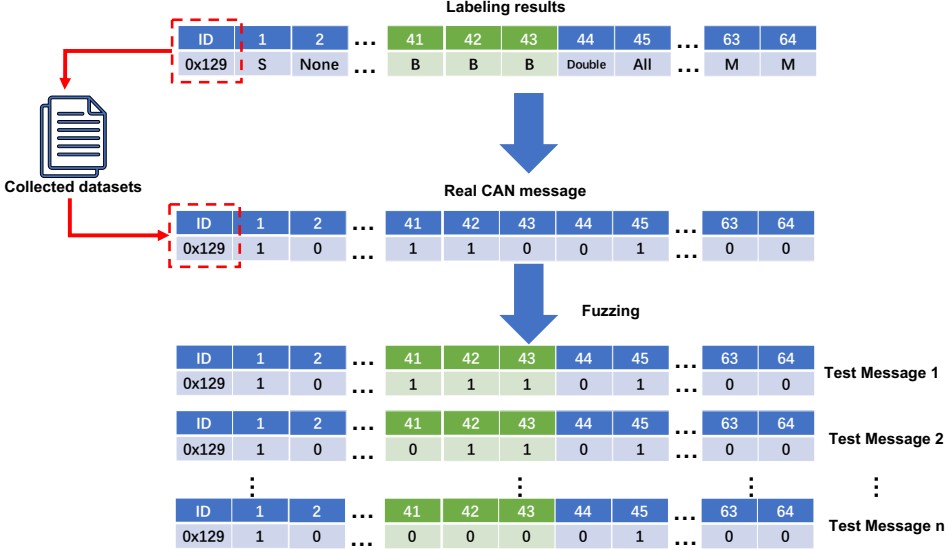

**Figure 6.** Test message generation process.

Although the fuzzed test messages can trigger most car body responses, there are still some car body behaviors that cannot be triggered by the test messages. The reason for this is that these body behaviors are controlled by the driver rather than by CAN messages, e.g., seat belts and vehicle lights, and CAN messages only characterize the state of the part. In order to obtain a description of the state of these messages, the packet analysis method is used to obtain candidate messages. In order to capture CAN messages containing descriptions of specific states of the body, specific operations are performed for short periods of time, with the CAN message recorded in the CAN network during these periods. Based on the screening and marking results obtained in the previous stage, the CAN traffic captured during a specific operation is screened, and the number of flips

is calculated based on the bit marked B. If the number of flips matches the number of operations, the marked bit characterizes the status of the part being operated. Since CAN messages usually use 1 to represent the dominant result, the message corresponding to the marker bit of 1 is recorded as a candidate message.

### 3.4. AND Bit-by-Bit

In the previous steps, the IDs related to the vehicle body behavior were filtered, and the bits related to the body in the corresponding messages were marked. Afterward, using fuzzified test messages and packet analysis, a large number of candidate messages that could trigger the body behavior and describe the vehicle behavior was obtained. In this phase, the bit-by-bit AND operation are used to quickly locate the bits in the candidate telegrams that are related to the control or state of a specific body behavior.

Based on the analysis of a large number of DBC files, the control bits in CAN messages with a value of 1 are usually valid, i.e., there is a body action response when the control bit is 1. Therefore, in the control and description messages corresponding to the collected vehicle behaviors, the valid bits controlling the behavior must all have a value of 1, while the bits not related to the behaviors usually do not have a value of 1. Based on this, in this method, a column AND bit-by-bit process are performed for the candidate messages while reversing the valuable bits. After this operation, the value of the AND is also 1 because all valid bits have a value of 1. For non-valid bits, the value obtained is 0 because these bits do not all have a value of 1. Therefore, AND is suitable for quickly locating the bits of the data field with a value of "1" from the candidate information that can trigger the body behavior and describe the vehicle behaviors. This method is faster and more accurate than manual screening and other logical operations. Based on this theory, the method described in this paper can quickly locate the valid bits that control specified body behaviors. This phase involves the following steps:

- First, all the bits in the candidate message that are not marked B are set to 0 to prevent other vehicle behaviors from interfering with the inverse result;
- After that, the values of the bits marked B in the data field of the candidate message undergo the column AND bit-by-bit process, and each data bit marked B is traversed. The control bits associated with the body operation have a value of 1, as shown for bits 41, 42, and 43 of each message in Figure 7. Unrelated bits have a value of 0; for example, bits 44 and 45 are shown in Figure 7;
- After this operation, this study obtained the valid bits that control or indicate the body behavior in the candidate messages.

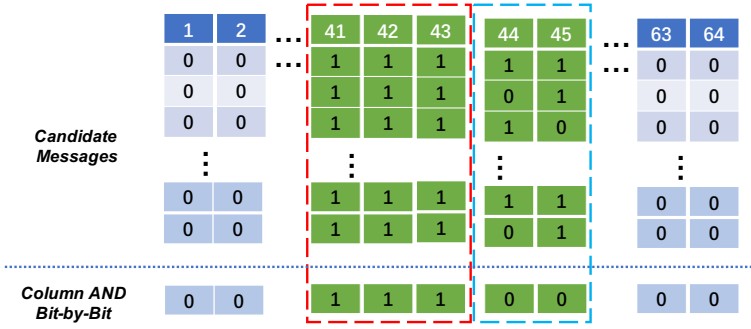

**Figure 7.** Column AND bit-by-bit.

## 4. Performance Evaluation

To evaluate the performance of the method proposed in this paper, it was implemented on an actual vehicle. The reversal was done to obtain the messages and specific bits that control or represent the car body's behaviors in CAN traffic. The algorithm's accuracy was evaluated with real applications based on existing DBC files using inverse results. In addi-

tion, we evaluated the execution performance of the framework. Finally, we investigated the algorithm's advantages for applications.

### 4.1. Performance in a Real Vehicle

In order to evaluate the inverse method proposed in this paper realistically and effectively, a mid-size sedan manufactured in 2017 was used as the test vehicle, and a representative internal network implementing the standard CAN protocol was employed. This means that the ECU in the CAN network could send and receive telegrams with a data field lengths of 64 bits. In addition, the DBC file for this vehicle was obtained [29] and used to evaluate the ground truth of our method. To obtain the experimental data, we used a combination cable to connect the vehicle's OBD-II interface to the USB interface of a laptop, and we used a script to write the data to a local log file. Following the experimental approach used in this paper, we collected the CAN traffic from the car when it was at a standstill ($\Omega_s$), at a standstill performing only body operations ($\Omega_b$), and during daily driving ($\Omega_m$) in the data collection phase. Quantitative descriptions of the datasets are given in Table 3.

**Table 3.** Datasets and message counts.

| Dataset Type | CAN Message Count |
|---|---|
| Standstill($\Omega_s$) | 159,890 |
| Only body operations($\Omega_b$) | 605,112 |
| Driving($\Omega_m$) | 132,707 |

The CAN traffic data collected from the test car under different conditions were first grouped by ID, and for each ID corresponding data field, the bit flip rate was calculated for the time series. The data fields corresponding to each ID were labeled based on the results of the bit-flip rate. In the actual experiment, we collected 81 IDs from the CAN traffic, of which 8 IDs were filtered and determined to be related to body behaviors based on the labeling results because their data fields contained only B-labeled messages.

For the filtered CAN message, fuzzing was performed on the bits in each message with only B tags, and the original data from the irrelevant bits were kept to generate the test messages. As shown in Table 4, eight body operations can be triggered by sending a test message. We recorded each message that triggered the body behavior and executed the AND bit-by-bit process. We executed the possible body operations five times for the behaviors that could not be triggered by the test message. We recorded the number of changes in the bits marked as B. A message with five changes was recorded, and the AND bit-by-bit process was executed for messages with a marked bit value of 1. Eventually, we obtained the CAN messages describing the specified automobile body behaviors in the CAN traffic and determined which bits in the data field controlled or indicated those body behaviors. The reverse results are shown in Figure 8. The results show that the CAN messages controlling the automobile body behaviors are concentrated in operations such as lights and wipers. The number of bits controlling these behaviors is small, usually 1–3 bits. Most of the other typical body behaviors, such as pedals, seat belts, and doors, are controlled by the driver, and CAN messages only describe their statuses. The number of data bits characterizing these vehicle behaviors was usually high, e.g., above 4 bits.

**Table 4.** Triggerable and non-triggerable body behaviors.

| Vehicle Body Operation Triggered by Testing | Non-Triggered Automotive Body Operations |
|---|---|
| Hazard Lights | Brake Pedal |
| Left Turn Signal | Accelerator Pedal |
| Right Turn Signal | Gear |
| Low Beam Lights | Driver's Seat Belt |
| High Beam Lights | Passenger's Seat Belt |
| Fog Lights | Driver's Door |
| Emergency Flashers | Passenger's Door |
| Trunk | Left Back Door |
|  | Right Back Door |

| Body-related Event | ID | Data Field (bits 1–64) |
|---|---|---|
| Hazard Lights | 0x091 | √ at bit 12 |
| Left Turn Signal | 0x091 | √ at bit 10 |
| Left Turn Signal | 0x09A | √ at bit 21 |
| Right Turn Signal | 0x091 | √ at bit 11 |
| Right Turn Signal | 0x09A | √ at bit 20 |
| Low Beam Lights | 0x09A | √ at bits 4, 5; √ at bit 30 |
| High Beam Lights | 0x09A | √ at bits 1, 2; √ at bit 30 |
| Fog Lights | 0x09A | √ at bit 3; √ at bit 30 |
| Emergency Flashers | 0x09A | √ at bits 1, 2 |
| Low Windshield Wipers | 0x09A | √ at bit 42 |
| High Windshield Wipers | 0x09A | √ at bit 41 |
| Brake Pedal | 0x165 | √ at bits 1, 2, 4, 8, 9 |
| Brake Pedal | 0x415 | √ at bits 10, 11 |
| Accelerator Pedal | 0x165 | √ at bits 22–27 |
| Accelerator Pedal | 0x420 | √ at bit 42 |
| Gear | 0x165 | √ at bits 57–61 |
| Gear | 0x228 | √ at bit 6; √ at bits 10, 11, 12; √ at bit 17; √ at bits 38, 39, 40, 41 |
| Driver's Seat Belt | 0x340 | √ at bit 34 |
| Passenger's Seat Belt | 0x340 | √ at bit 33 |
| Driver's Door | 0x43E | √ at bit 37 |
| Passenger's Door | 0x43E | √ at bit 38 |
| Left Back Door | 0x43E | √ at bit 39 |
| Right Back Door | 0x43E | √ at bit 40 |
| All Doors Closed | 0x43E | √ at bit 28 |
| Trunk | 0x43E | √ at bit 43 |

**Figure 8.** Reverse results of vehicle body-related CAN messages.

### 4.2. Accuracy

To assess the accuracy of the methods, we compared the reverse results with the definitions contained in the DBC file for each data bit in each ID. Since this DBC file is specific to this range of models and some features are not optional for our test vehicle, we artificially filtered the defined information in the DBC file. By analyzing and filtering the 82 IDs contained in the DBC file, giving a total of 5248 bits, we filtered and found that 51 bits were related to body operations. The rest represent kinematic-related bits, constant heartbeat packet bits, and many CTR and checksum bits.

Since the reverse body-related CAN messages involved in this study are similar to those used in the classification problem, three metrics were used to evaluate their performance, accuracy, precision, and recall, the values of which were calculated using Equations (3)–(5), respectively. We recorded true positives (TP) as those bits defined in the DBC file as controlling or indicating body behaviors and whose reverse results were consistent with that definition. Bits in the DBC file specified as unrelated to body behaviors and whose results excluded them were recorded as true negatives (TN). Bits in the DBC file defined as irrelevant to the body but whose reverse result was related to the body were recorded as false positives (FP). Bits in the DBC file specified as being related to the body but whose inverse result was not related to the body were recorded as false negatives (FN).

$$\text{Accuracy} = \frac{\text{TP} + \text{TN}}{\text{TP} + \text{TN} + \text{FP} + \text{FN}} \tag{3}$$

$$\text{Precision} = \frac{\text{TP}}{\text{TP} + \text{FP}} \tag{4}$$

$$\text{Recall} = \frac{\text{TP}}{\text{TP} + \text{FN}} \tag{5}$$

After counting, the 1340 bits were classified as TNs, 45 bits were TPs, 3 bits were FPs, and 6 bits were FNs. The method proposed in this paper was shown to have 99.35% accuracy, 93.75% precision, and 88.24% recall. The main reason for a precision value of

only 93.75% is that some bits indicated that the stationary state of the car changed with the use of the brake or gas pedal, so these bits were classified as brake and gas pedal events. The recall rate was only 88.24% because we used a home car as the test vehicle, and we could not carry out some body-related events, such as the opening of airbags. Although the above reasons caused decreases in the accuracy and recall rate, the reverse method based on marker bits proposed in this paper was shown to have substantially improved efficiency and accuracy values than the traditional manual reverse method.

*4.3. Strengths and Efficiency*

4.3.1. Strengths

In this paper, we achieve bit-level inversion of CAN messages related to body behaviors based on the labeled bits of data fields, and the results showed an accuracy level of 99.35%. In the experiments, in addition to the test messages generated using the fuzzing scheme, the packet analysis method was employed to compensate for the reverse results of vehicle behaviors that could not be triggered by the test messages. In addition, the AND bit-by-bit method was used for the valid bits in the final location. On the one hand, this improved the accuracy compared with manual inversion. On the other hand, it does not require a high number of test messages and thus increases the speed of inversion.

To translate those operations that fuzzing could not identify, we performed a packet analysis of the subject's operation. In the case of braking, for example, the fuzzing method does not work due to the checksum in the data field. Without any other operation, we captured the CAN bus data from the test vehicle, which had brakes that had never been applied to the stepped state. Through the packet analysis, we obtained two sets of valid information. One group had an ID of 0x165 with the corresponding bits 1, 2, 4, 8, and 9, and the other had an ID of 0x415 with the corresponding bits 11 and 12. Considering the CAN bus's arbitration mechanism, we know that ID 0x165 corresponds to the control signal of the brake, while ID 0x415 indicates the status of the brake.

When undergoing the AND bit-by-bit process, the method described in this paper can translate vehicle body behaviors using just 10 candidate messages. In the case that the number of candidate messages is $n$, if the probability of any bit in the data field is set to 1 is $p$, the probability of the irrelevant bit AND bit-by-bit resulting in 0 is $1 - p^n$. Then the correct rate at this point is $(1 - p^n)^m$ (where $m$ is the number of irrelevant bits). During our experiments, we calculated that the maximum number of potential body data bits in the data field of 12. However, a more extreme value of 20 bits was used to verify the accuracy, and it was assumed that only 15 bits were valid (which also means that 5 bits are actually not body-related). Since these are independent, we considered the value of P to be 0.5. Figure 9 showed an accuracy of 99.5% when $n$ was 10 for the assumed extreme case. In actual vehicle experiments, the bit-tagging method in this paper would not result in $m$ being greater than 5, so the accuracy would be higher in the actual case.

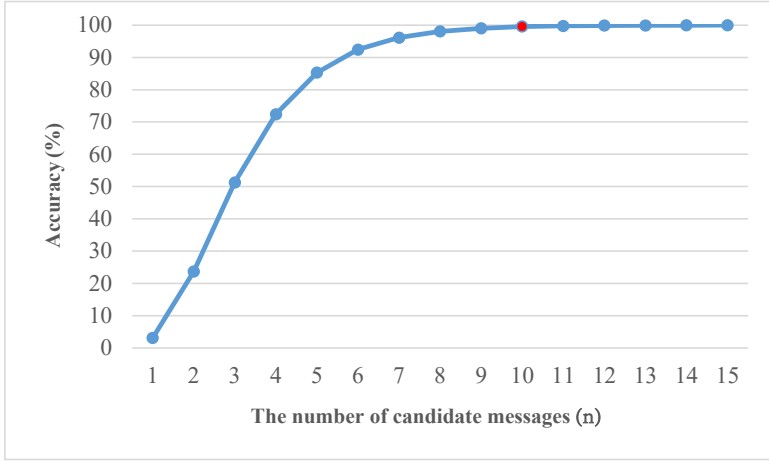

**Figure 9.** Accuracy versus candidate messages.

### 4.3.2. Efficiency

In order to evaluate the time consumption of the inverse process in this paper, we performed time statistics for the whole process. Since some parts involved in the inverse process need to be performed manually, we only retrieved accurate time statistics for the data processing and analysis part, as well as rough estimates for the manual parts. The experiment was divided into four parts: data collection, labeling and filtering, fuzzing and packet analysis, and bit localization. Data processing and analysis were performed on a 64-bit CentOS 7.4.1708 server using python 3.7.1. The server had 8G RAM and the Intel(R) Xeon(R) Gold 6248 CPU @ 2.50GHz with four cores and four threads. All code was run in a single thread, and the final efficiency in terms of time consumption is shown in Table 5. The total time taken to retrieve the body-related messages from the reverse experimental vehicle was 4087.85 s.

**Table 5.** Time consumption of the reverse method.

| Step | Time Cost (s) |
| --- | --- |
| Data Collection [a] | 1380 |
| Labeling and Filtering [b] | 97.85 |
| Fuzzing [b] | 2460 |
| Packet Analysis [a,b] | 150 |
| AND Bit-by-Bit [b] | 0.00001 |

[a] Performed manually; [b] Performed automatically.

### 4.4. Application and Discussion

#### 4.4.1. Application

The translation method proposed in this paper for vehicle body-related messages based on labeled bits can be applied to almost all commercially available vehicles. The method described in this paper only needs to collect CAN traffic and send test messages through the OBD-II interface. All vehicles must be equipped with the OBD-II interface [30] as a common automotive standard before they are launched [31–34], because our method can use the OBD-II interface for data collection and sending for any vehicle model. In addition, OBD-II connection devices are now readily available in the market, ranging in price from tens [35] to hundreds of dollars [36], all of which can be easily used to send and receive CAN messages. Therefore, the method proposed in this paper can be easily applied to most car models.

#### 4.4.2. Contribution to Automotive IDS

The CAN message translation method proposed in this paper can assist with research on automotive intrusion detection systems (IDS). The CAN intrusion detection system was proposed to detect anomalies by analyzing CAN traffic [37–41]. However, studies on this system were based on message transmission characteristics, such as periodicity, entropy [38], and remote frames [37], which are irrelevant to the behavior and state of the vehicle. These methods are almost ineffective when an attacker sends attack messages with transmission characteristics that are consistent with those of regular messages. The method proposed in this paper can help researchers quickly obtain the specific meaning contained in CAN messages so that they can use the message contents to design more direct and effective intrusion detection rules, e.g., no control commands for stalling or braking should appear while the vehicle is driving normally and no doors should appear open when the vehicle is accelerating. Therefore, IDS researchers and designers can use the results obtained with the method described in this paper to design content-based intrusion detection systems that can identify attacks with greater efficiency than existing methods. In addition, the content-based approach is faster and more applicable than IDSs, which utilize machine learning algorithms.

### 4.4.3. Discussion

In this study, we proposed a reverse method for body-related CAN messages based on tagged bits. This method collects CAN traffic from different vehicle states and tags the data fields of the message based on these states. The messages are filtered, fuzzed, and packet analyzed based on the tagging results. Finally, the bits that control or characterize body behaviors are located using the column AND bit-by-bit operation. Thus, the definition of body behaviors is obtained from the DBC file. In actual vehicle experiments, our method was shown to have high reverse accuracy. The lack of perfection in the results is because the car body cannot be triggered or interfered with by other information. The packet analysis process used in this method enables the reverse results to cover more body behaviors, and the column AND bit-by-bit process improve the accuracy and speed. Finally, our approach can help researchers in automotive IDS to design more efficient content-based intrusion detection methods. In terms of applications, the process can be applied to CAN information inversion work for all vehicle models.

### 5. Conclusions

In this paper, a reverse method for body-related CAN messages based on labeled bits was proposed. This method tags each bit of the data field according to the different states of the vehicle. Based on the results of the bit labeling, messages related to the body were filtered out, and fuzzy messages and packet analysis were combined to discover candidate messages. AND bit-by-bit quickly located the bits in the CAN message that can control or characterize specific body operations. The inverse method achieved an accuracy level of 99.35%. In addition, the method utilized only the OBD-II interface to send and receive data, a process that can be model-independent. Our method could help automotive security researchers quickly obtain body-related control CAN message or CAN status messages without DBC files which, in turn, would allow them to propose intrusion detection schemes based on message content. Finally, the results of the inverse method proposed in this paper could be used to advance the automotive IDS research field and could also be equally valuable for research on autonomous driving, implicit authentication, and other related fields.

**Author Contributions:** Methodology, Z.B.; Software, S.Z.; Formal analysis, C.W.; Investigation, Z.B.; Data curation, S.Z.; Writing—review & editing, C.W.; Supervision, G.X. (Guosheng Xu); Project administration, G.X. (Guosheng Xu) and G.X. (Guoai Xu); Funding acquisition, G.X. (Guoai Xu). All authors have read and agreed to the published version of the manuscript.

**Funding:** This work is supported by the National Key Research and Development Program of China under Grant No.2021YFB3101500, China Postdoctoral Science Foundation under Grant No.2021T140074, and the National Natural Science Foundation of China under grant No.62102042.

**Data Availability Statement:** The data supporting this study's findings are available from the corresponding author upon reasonable request.

**Acknowledgments:** The authors would like to thank the editors and all the reviewers for their valuable comments.

**Conflicts of Interest:** The authors declare no conflict of interest.

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
