# Peer review of "A Method for Translating Automotive Body-Related CAN Messages Based on Labeled Bits"

_applsci, doi:10.3390/app13031942_

Round 1

Reviewer 1 Report

1. The article describes the method to decode the CAN signals of the car body, which is valuable for engineering practice, but lacks theoretical analysis.

2. The method described in the article mainly aims at some simple signals in the body CAN, such as the light status, and lacks verification of some continuous variables. 

3. The same signal may appear in different IDs. How to identify it using the method in this article?

Author Response

Thank you for your constructive comments on the paper. We have carefully studied all the review comments and have made every effort to revise our manuscript. Please see the attachment for the response to the review report.

Reviewer 2 Report

1. This article is actually not about protecting the ECU from cyberattack (IDS), but how to reveal the bit location that controls a car. The contribution of the article to IDS must be improved.

2. The theory basis for your proposed method are not enough. Why did you use longitudinal terminology rather than column one? Why do you use AND rather than OR? It needs explanations.

3. Why is the fusion result in column put in three different datasets (S/M/B or M/B)? Logically, each column should be represented by one dataset.

4. What is the fusion method you used and how does it works?

5. What does the meaning of “0”, “S/M/B”, and “M/B” results? There should be only one result of fusion not more than one.

6. What is bit-flip rate and why did use this mechanism? 

7. What is DATA field and what does it contain?

8. The accuracy formula comes from nowhere in this article, and why “5”? Need explanations.

9. There is no explanation how many bits used in ECU for your case. Is it 64 bit or more or less?

10. The labelling mechanism shown in Figure 6 needs more explanation. How can be different bit values labelled with the same label?

Author Response

(The authors gave the same response as above.)

Round 2

Reviewer 1 Report

The authors can describe the relationship between their previous work and this paper in section 2.

Author Response

Thanks for your suggestion. Following your suggestion, we added the relationship between the previous work and this paper in section 2, as detailed in lines 164-179.

Reviewer 2 Report

Line 386, where did you get the value of 5 in (1-p^n)^5? It's not explained yet.

Author Response

We apologize for the confusion caused by the lack of explanation of value 5. To reduce confusion, we added a parameterization of the formula for calculating the correct rate in line 399. We have assumed an extreme case in lines 401-403: assuming that 20 bits in the data field are marked as potentially body-related and that 15 bits are valid, this also means that 5 bits are not body-related. The value 5 is derived from this assumption. An explanation of this value is given in lines 402-403 of the article for greater clarity.